# Immunogenic Effects of *Dietary Terminalia arjuna* Bark Powder in *Labeo rohita*, a Fish Model: Elucidated by an Integrated Biomarker Response Approach

**DOI:** 10.3390/ani13010039

**Published:** 2022-12-22

**Authors:** Dharmendra Kumar Meena, Soumya Prasad Panda, Amiya Kumar Sahoo, Prem Prakash Srivastava, Narottam Prasad Sahu, Mala Kumari, Smruti Samantaray, Simanku Borah, Basanta Kumar Das

**Affiliations:** 1ICAR—Central Inland Fisheries Research Institute, Barrackpore 700120, India; 2Central Institute of Fisheries Education, Mumbai 400061, India

**Keywords:** circular bioeconomy, integrated biomarker response approach, *Terminalia arjuna*, green food products

## Abstract

**Simple Summary:**

It is widely believed that aquaculture has the greatest potential to provide both economic and nutritional security. Bacterial infections in the culture system result from intensified aquaculture methods, resulting in a significant economic loss for fishermen. Because antibiotics are the standard treatment for bacterial infections in Southeastern countries, antimicrobial resistance has emerged as a major health concern. Herbal remedies are being promoted as a replacement for the undesired medicines. In Ayurveda, the *Terminalia arjuna* plant is considered a “miracle plant” since it has been used for centuries to treat serious illnesses in humans. However, applications in aquaculture remain in their early stage. The purpose of this study is to assess the therapeutic value of incorporating powdered *Terminalia arjuna* bark with *Labeo rohita*. Based on the findings, the researchers suggest adding *Terminalia arjuna* bark powder to fish food at a rate of 12.3 g·kg^−1^ (as determined by the broken line regression equation) to help the fish become resistant to disease.

**Abstract:**

Utilizing agro-industrial waste and herbal products to create a circular bioeconomy is becoming increasingly popular. *Terminalia arjuna* is a significant ethnomedicinal plant that has not yet been exploited in animal feed. In the present study, nutritional *Terminalia arjuna* bark powder-based fish feed was created and supplied to a candidate fish species *Labeo rohita* at varied levels: 0% (0 g/kg), 0.5% (5 g/kg), 1% (10 g/kg), and 1.5% (15 g/kg). These treatment groups are denoted as CT, T1, T2, and T3, respectively. Utilizing a contemporary comprehensive biomarker response strategy, the study clarified the genomic influence of dietary herb inclusion. In response to bacterial infection, the immunogenic genes, STAT 1 (signal transducer and activator of transcription 1), ISG 15 (interferon stimulating gene), and Mx “myxovirus resistance gene”, were shown to be elevated. The results of densitometry demonstrated a dose-dependent increase in STAT 1 and ISG 15, with Mx exhibiting maximal values at 1 g/kg TABP (*Terminalia arjuna* bark powder-based feed). This is the first study to identify TABP as an immunomodulator in fish and established the IBR (Integrated Bio-marker Response) as a reliable marker in evaluating the impact of multiple drivers in a holistic manner. Thus, the present study cleared the path for TABP to be utilized as an effective feed additive which enhances the specific adaptive immune system of the fish for the production of the Green fish product for a sustainable circular bioeconomy.

## 1. Introduction

Aquaculture is one of the fastest growing food production systems, providing nutrition and livelihood security to a large population in India and around the world. The Indian aquaculture revolves around the six species, including three Indian major carp (IMC), contributing around 60–70% of total production. However, this production trend is not enough to meet the ever-increasing demand in the country. The aquaculture revolves around the Indian major carp rohu (*Labeo rohita*), catla (*Gebilion catla*), and mrigal (*Cirrhinus mrigala*) in monoculture and polyculture systems with supplementary or complete feeding.

Among the IMC, the culture of *Labeo rohita* has gained importance in carp polyculture due to its adaptability with other species as a column feeder, better growth, and consumer preferences. Every step towards intensification invites significant disease occurrence, mainly due to bacteria, followed by viruses, parasites, etc. The most prevalent bacterial pathogens are *Edwardsiella tarda*, *Aeromonas hydrophila*, *Flavobacterium* spp., etc., and the pathogenic bacteria are supposed to invade the aquaculture system more rapidly in the years to come. Primitive vertebrates (agnathans, sometimes known as jawless fish) were the first to combine innate and adaptive reactions. Thus, hagfish and lampreys employ LRRs as variable lymphocyte receptors, whereas higher vertebrates (cartilaginous and bony fishes, amphibians, reptiles, birds, and mammals) evolved the major histocompatibility complex, T-cell receptors, and B-cell receptors (immunoglobulins) as adaptive weapons to augment innate responses. Fish possess extensive cytokine networks, whereas invertebrates possess signal molecules with similar structures. Adaptive immunity was so effective due to its high specificity, antibody maturation, immunological memory, and secondary reactions that it allowed higher vertebrates to limit the number of variants of innate molecules originating from both invertebrates and lower vertebrates [1]. In the mammalian system, a varied range of cellular pathways is being activated upon infection by viruses, leading to the release of nearly 200 interferon-stimulated genes (ISGs). In most cases, several genes mediate the antiviral response, but sometimes a single gene also regulates it. The Mx is an antiviral protein belonging to the family of dynamins that includes an amino-terminal with a G-domain, an interactive central area, and a leucine zipper as an effector area of GED or GTPase [2]. The STAT1 (Signal Transducer and Activator of Transcription) is an essential signal transduction protein that is intricate in the interferon pathway. The STAT1 plays a significant role in the non-specific defense system [3]. The ISG15 is an interferon-stimulating antiviral gene. Although ISG15 was thought to be an integral part of classically antiviral immunity, it has newly appeared as a regulator of genome steadiness, with main roles in the DNA nicking inhibition to modulate p53 signaling and error-free DNA replication [4]. Medicinal plants, herbs, herbal extracts, and their associated products are well known for possessing various health-beneficial properties, and since ancient times, they have been used to treat various geriatric and chronic diseases in human beings. Medicinal plants and their product have also been used in aquaculture as a growth enhancer, immunomodulator, and nutraceutical. Initially, medicinal plants are believed to trigger only the non-specific immune system of fish; however, previous research has highlighted their role as adaptive immune system-enhancers of fish as well [5]. A comparable multiple-herb approach recognized as “Fufang” is a vital constituent in conventional Chinese medicament and is applied to attain improved therapeutic outcomes and decrease side effects and herbal toxicity [6,7]. Tan et al. (2020) [8] studied the impacts of dietary *Ginko bibola* leaf (GBE) extracts (0.0–10.0 g·kg^−1^ of feed) on the growth performances and expression of immunogenic genes in the hybrid group (*Epinephelus lanceolatus*♂ × *Epinephelus fuscoguttatus*♀).

Vaccination is an expensive and practically unfeasible strategy as compared with other modes of treatment for bacterial diseases, specially when the cultured species has significant commercial value; therefore, it is not preferably adopted as economically viable. The Southeastern Asian countries are mainly depending on carp species or Chinese carp, and for those species, vaccines may not be an economically viable and practically feasible option. In the recent past, fish species have been affected by viral infection, as well as bacterial, parasitic, and other microorganisms [9].

The non-specific defense system of fish is very important against microorganisms including bacterial pathogens; therefore, using organic, inorganic, or synthetic matters, nutraceuticals and herbal extracts as immune stimulants have been used as better alternatives to antibiotics to control fish diseases [10,11]. In addition, some of the genes, such as Mx, are occurring as natural gate keepers of the fish immune system that protect the fish from pathogens during early stages and may get activated due to an external inducing agent. However, the mechanism of action behind the activation of Mx gene expression due to herbal products is still not clear.

The *Terminalia arjuna* has been reported to possess excellent antioxidant and antimicrobial properties and has been proved to be safe for host fish species [12,13,14,15]. However, the effects of *Terminalia arjuna* on the specific immune system have not been established so far. The majority of industrial processes may be summed up in three steps: “take, make, dispose” (using, creating, and discarding), since they all entail the same limited resources, generate waste, and produce goods that are eventually discarded. *Terminalia arjuna* is a significant ethnomedicinal plant; however, it has not been employed per se from Green bioeconomy viewpoints. Several biomarkers are counted as useful indices in delineating growth monitoring and risk assessment in response to varying dietary additives and the changing quality of the culture system. Nonetheless, the observed data using the biomarker-based monitoring approach are often limited and hard to interpret due to the lack of an integrated statistical analysis. The application of an integrated biomarker response (IBR) index enables a holistic measurement of the stress levels faced by the fish. This index has been quite successfully applied in fishes for evaluating the effect of environmental pollutants and chemicals [16,17]. Debbarma et al. (2021) [18] have also evaluated the effects of biofloc system parameters on the welfare status of *Ompok bimaculatus*. However, it has not used as such to evaluate the immunogenic effects of herbal feed in fish species. In this backdrop, the present study was an attempt to evaluate the potency of dietary *Terminalia arjuna* bark powder-based feed on immunogenic genes using an integrated biomarker response approach in fish, *Labeo rohita.*

## 2. Material and Methods

### 2.1. Experimental Design, Setup Feeding Trial, and Sample Collection

The study follows a completely randomized design for the feed trial and challenge study. The study was conducted in triplicate of the treatments. A 90-day-long indoor feed trial was followed by a challenged study to evaluate the expression of three immunogenic genes: STAT1, ISG15, and Mx. The experimental setup includes four groups based on *Terminalia arjuna* bark powder inclusion in fish feed, CT, 0% inclusion (TABP0), T1, 0.1% inclusion (TABP01), T2, 1% inclusion (TABP10), and T3, 1.5% inclusion (TABP15). Five hundred and forty fish (average weight, 20.7 ± 0.34 g) were distributed into four dietary treatments in triplicate. The rearing size of the fish was selected because, from this stage onward, they are being used as stocking materials and are transferred to the new environments. Therefore, there is very much a possibility of infection occurring. So, in view of this, the size of the fish was selected. The area of the cultured fish is kept in the upper side, it was taken as 10 L for one fish, and the total volume is 450 L in a 500 L capacity tank. The fish were reared in a 500 L tank of a flow-through system and fed twice daily (9:30 a.m. and 5:00 p.m.) up to satiation. The experiment was conducted for three months (90 days) as a cultured period for the rearing stage of fish and to provide a suitable time to reflect the dietary effects of TABP on the fish specific immune system, unlike other herbal materials. Further, this study was followed by a challenge study with two pathogenic bacterial isolates (*Edwardsiella tarda* and *Aeromonas hydrophila*) that lasted up to 10 days. The pathogenic bacterial strains were grown for 24 h in a BOD incubator at 30 °C on tryptone soya broth (TSB). The cells were collected and washed thrice in sterile PBS before being put into PBS at a concentration of 1.97 × 108 cells/mL. Each fish was challenged with 100 μL of bacterial suspension, which corresponded to 105 cells/mL. The relative survival percentages (%) in the challenged fish and the changes in the behavioral morphology were observed for up to 10 days. At the end of the feeding trial and challenge study with two bacteria, the samples were collected and preserved in RNAlater. The CRD experimental design was followed to describe the means equality of dependent variables in two or more treatments.

### 2.2. Total RNA Extraction, RNA Quantification, and cDNA Transcription

Total RNA was extracted from *L. rohita* brain tissue using an RNA isolation kit from the Quigen Rnase mini kit, as directed by the manufacturer. Based on 1% agarose gel, two notable bands at 28s and 18s were evaluated in terms of the quality of the RNA following RNA extraction. Spectrophotometric verification of RNA purity and quantity was performed using a NanoDrop ND-Bio-Rad, USA). To ensure gDNA-free samples, Reverse transcriptase-deficient reactions (RT min) were carried out. Using the Super Script III first-strand synthesis super mix for qRT-PCR, about one microgram of total RNA was reverse-transcribed for preparing cDNA (Invitrogen, Waltham, MA, USA).

### 2.3. cDNA Synthesis and Amplification

PolyA^+^ RNA was reverse-transcribed to cDNA using oligo dT 18–20 primers. Five micrograms of total RNA were mixed with oligo dT 18–20 primers (NEB England, Hitchin, UK), made up to 10 μL, heated at 65 °C for 10 min, and then immediately chilled in ice for 5 min. Samples were adjusted to a final volume of 20 μL by the addition of an RT buffer, M-MLV reverse transcriptase 5U lL-1, 100 mM dNTP, and RNase inhibitor, kept at 37 °C for 60 min, and then stored at −20 °C before PCR.

### 2.4. Primer Designing, Semi-Quantitative PCR of Immunogenic Genes, and Densitometric Analysis

Primers were designed using Primer 3 software. Semi-qPCR PCR was performed by taking the cDNA sample, using specific primers and the housekeeping gene β-actin. The details of the primers used in the present study are given in Table 1.

### 2.5. Semi-Quantitative PCR

The polymerase chain reaction conditions for the Mx gene are as follows: initial denaturation for 5 min at 94 °C, final denaturation of 35 cycles of 15 s at 94 °C, annealing for 30 s at 56–65 °C, elongation for 45 s at 72 °C, and final elongation for 7 min at 72 °C. For ISG 15 and STAT 1, the polymerase chain reaction conditions are as follows: initial denaturation for 5 min at 94 °C, final denaturation of 40 cycles of 15 s at 94 °C, annealing for 30 s at 68 °C, elongation for 45 s at 72 °C, and final elongation for 5 min at 72 °C.

### 2.6. Densiometric Analysis, Networking, and Correlation among Densiometric Parameters and Treatments

The amplified genes were imaged in a gel doc system, and densitometric analysis, including gel quantification and characterization, was performed in LAB IMAGE software (Bio-Rad). Overall association patterns, significance, and networking among genes, treatments, and densitometry parameters were established using PAST, Minitab 18 software https://softwarerequest.psu.edu/, accessed on 9 December 2021).

### 2.7. Gene Expression in qPCR

Immune genes were deployed to evaluate the expression pattern in *Labeo rohita* by RT-PCR (qPCR). The Light Cycler 480 I master mix was used for qPCR (Roche, Munich, Germany). Amplification was carried out using an Mx specific primer, and the details of the primer are given in Table 1. The amplicon sizes of Mx, ISG15, STAT1, and β-actin are 165, 200,146, and 265 bps, respectively.

The Primer efficacy output was checked based upon the standard curve line slope and a melting curve. Approximately 1.0 μL of 10-fold diluted cDNA was mixed with 0.5 μL (5 pmol) of each primer (the Mx forward primer and reverse primer (10 μL)), 5 μL of 2x I master mix on the Light Cycler 480 SYBR Green (Roche, Munich, Germany), and 3 μL of nuclease-free to a final volume of 10 μL. The qPCR amplification was performed in triplicate, with the β-actin (housekeeping gene) as the control, under the following conditions: 95 °C for 10 min, followed by 40 cycles of denaturation at 95 °C for 10 s, amplification at 55 °C for 10 s, and extension at 72 °C for 10 s, and the PCR conditions for the three genes were the same as those in the case of semi qPCR.

### 2.8. qPCR Analysis

The 2^−ΔΔCT^ technique was used to calculate the relative gene expression of the targeted genes in comparison to the reference gene (β-actin). The quantification cycle (cq) values for each gene were calculated using Light Cycle SW 1.1 software and a second derivative, and the maximal method was used for absolute quantification. When the efficiency was ~100%, 2^−ΔΔCq^ fold was calculated by method [19], taking the β-actin relative gene expression. The 2^−ΔΔCq^ was used to compute the fold expression for each sample in relation to the calibrator. Each group’s average (in triplicate) folding expression was derived and presented as a median value. The 1.5% agarose gel was used to measure the desired length of the band for 8 μL of the qRT-PCR product.

### 2.9. Integrated Biomarker Approach for Elucidating the Effect of Dietary TABP on Immunogenic Genes and Treatments

To assess the multi-biomarker response in *L. rohita*, we computed integrated biomarker responses (IBRs) for several biomarkers in immunogenic genes and displayed corresponding star plots. In accordance with the proposed approach, we carried out our investigation [20,21]. The scores (S) of all biomarkers assessed in a specific treatment and genes were represented by star plots, and the following formulas were used to generate IBRs:Ai = Si/2 sinβ (Si cosβ + Si + 1 sinβ) Ai = Si/2 sinβ (Si cosβ + Si + 1 sinβ)
where β  =  Arc tan (Si  +  1 sinα/ Si  −  Si  +  1 cosα) and α  =  2π/ n, Sn  +  1  =  S1

There is a total of n biomarkers utilized in the computations; therefore, Ai is the area that connects the two scores (S), and Si and Si + 1 are the two successive clockwise scores (radius coordinates) of a specific star plot. To compute the average value of each biomarker, the IBR index for each treatment was standardized to account for various genes.

### 2.10. Statistical Analysis

The data were analyzed in Microsoft Excel v.16, and significance was established by deploying SPSS 20. The images were edited in Paint 3D v.16.

## 3. Results

### 3.1. RNA Extraction and Quantification

For gene expression studies, RNA was isolated using the Qiagen RNA isolation Kit. Isolated RNA was separated on 1% agarose gel, and both 28s and 18s RNA bands were seen. The RNA concentration was 1500–1800 ng/μL.

### 3.2. Semiquantitative PCR

Two distinct bands were observed at 200 bp and 265 bp for β-actin (as the housekeeping gene) and Mx, respectively (Figure 1a). Semi-quantitative analyses showed two distinct bands at 200 bp and 265 bp for β-actin (as the housekeeping gene) and ISG15, respectively (Figure 1b). Two distinct bands were observed at 200 bp and 700 bp for β-actin (as the housekeeping gene) and STAT1, respectively (Figure 1c).

### 3.3. Densiometric Analyses and Correlation

The densiometric analysis parameters, such as the relative front (Rf), relative quantity (RQ), Lane (%), and Band (%), were measured. Overall, irrespective of any genes and treatments, the housekeeping gene β -actin showed a maximum value for these parameters. The values of Rf, RQ (ng), Band (%), and Lane (%) were 0.94–0.99, 1.33–149, 50.8–59.49, and 31.77–39.67, respectively (Table 2, Table 3 and Table 4). The order of densitometry was represented as follows: ISG 15 > Mx > STAT 1. In Mx densitometry parameters, the Rf showed that CT.90 differed significantly (*p* < 0.05) from the other treatment groups. The Rf showed a total of three groups based on significance values. RQ and Band (%) showed five groups, while Lane (%) revealed six groups based on significance values. In ISG 15, minimum Rf, RQ, Lane (%), and Band (%) values were recorded for T1.90, and maximum values were recorded in T2.100Ah. In ISG 15, Rf, RQ, Band (%), and Lane (%) revealed six, seven, four, and six groups based on significance. In STAT 1, minimum Rf, RQ, Lane (%), and Band (%) values were recorded for T1.90, and maximum values were recorded in T2. In 100Ah, Rf, RQ, Lane (%), and Band (%) exhibited six, four, seven, and seven, respectively. The mention-worthy point is that, in the feed trial, the maximum values for all parameters were ascertained by T3.90 in Mx and STAT1, while in the case of ISG 15, the maximum value was recorded in T2.90.

### 3.4. Correlation

The correlation matrix of the densiometric parameters of the genes is shown in Figure 2a–c.

The correlation values vary between −1 and 1, as indicated by the scale in the right-hand side of Figure 2a. The circle indicates the value of the corresponding parameters with the significance level of the degree of association.

The correlation values vary between −1 and 1, as indicated by the scale in the right-hand side of Figure 2b. The circle indicates the value of the corresponding parameters with the significance level of the degree of association.

The correlation values vary between −1 and 1, as indicated by the scale in the right-hand side of Figure 2c. The circle indicates the value of the corresponding parameters with the significance level of the degree of association.

STAT 1 showed a positive correlation with each other. Rf showed a maximum association with RQ, while RQ and band percentage showed a maximum correlation with Lane percentage.

The correlation matrix of the densiometric parameters of ISG 15 1 showed a positive correlation with each other. Rf showed a maximum association with RQ, while RQ and band percentage showed a maximum correlation with Lane percentage. Noticeably, Lane percentage showed a similar intensity of association with Rf, RQ, and band percentage. Band percentage showed a similar association with Rf and RQ.

The correlation matrix of the densiometric parameters of Mx was found to be positively correlated with each other. Rf showed a maximum association with RQ, while RQ and band percentage showed a maximum correlation with Lane percentage. Noticeably, Lane percentage showed a maximum association with RQ and band percentage. Band percentage showed the highest association with Lane percentage.

### 3.5. Networking

The network plot is based on its depiction by connecting edges and nodes. The present network plot is reflecting the correlation matrix between the treatments based on the expression level of the densiometric parameters of three immunogenic genes (Figure 3a–c).

The network plot of the Mx gene is showing that the reference gene Beta-actin is showing more connecting edges and similar nodes to T3.90, CT.Ah, and CT.90, while T3.90, T2.Et, and T1.Et showed a similar type of association. At the top of the network, CT90, T290, and CT.Et showed a similar area of nodes. The treatments such as B-actin, T2.Ah, T2.90, and T2.Et created a central rectangular network. The maximum number of connecting edges (14) and the area of the node are reflected by T2.Et, followed by T1.Et, which is showing 12 connecting edges but a greater area of the node as compared with T2.ET. T1.90 and T3.Ah showed a similar number of connecting edges (5) and the same area of the node. Overall, T1.90, T3.Ah, and T3.Et showed distinct distributions compared to their treatments, 90-day feeding trial, *A. hydrophila* infection, and *E. tarda* infection, respectively. B-actin takes a central position for T3.90, CT.Ah, CT 90, T2.90, T2.Ah, and T2.Et.

The networking of treatments of the STAT1 gene shows that the plot is bifurcated in approximately equal parts by the centrally distributed T1.Et, T2.Et, and T3.90. The reference gene B-actin is not in the central loop and it is connected by seven edges to T1.Et and T2.90. The CT.Et, T1.90, and CT.Ah did not connect to their treatment groups. Two groups, T1.Et, T2.Et, and T3.Et and T1.Ah, T2.Ah, and T3.Ah, showed a distinct triangle, while the group CT.90, T2.90, and T3.90 was interrupted by CT.Ah treatment. The area of the node of B-action is apparently similar to T2.90, CT 90, CT.Et, T3.90, T2.Ah, T1.Ah, T3.Et, T1.90, and T3.Ah. The T3.90 is connected by maximum overlapping edges (13), followed by both T2.Et and T1.Et (12).

The network of ISG 15 showed a mixed manner of both Mx and STAT 1 genes. The Ah treatment groups are highly scattered and do not form any distinct triangle of association with its subgroups. The triangle formed by T2.100Ah, CT.100Ah, and T1.100Ah is interrupted by B-actin and T2.90. Apparently, a similar nodes area is shown by T1.Et, CT.100Et, T2.100Et, and CT.90. The T2.100Et, T3.100Et, and CT.100Et and T2.100Et, T3., and T1.Et showed two distinct triangles of association. CT.90 has taken the central position of association, which is in a close relationship with T1.100Et, CT.100Et, and T2.100Et. The treatment groups of Ah are distantly distributed.

### 3.6. qPCR Analyses

The relative qPCR analyses showed varied expression levels (Figure 4). The Mx levels during infection varied considerably between the treatments (*p* > 0.05). Within the treatments, the Mx value increased considerably (*p* < 0.05) after Et infection but exhibited no significance after Ah infection (*p* > 0.05). However, between 90 days and Et infection, the Mx value grew considerably (*p* > 0.05) in T1, T2, and T3, but there was no significance in T3 (*p* > 0.05). The treatment effects on STAT1 expression were staggered. CT and T1, as well as T2 and T3, exhibited no statistical significance (*p* > 0.05). T3, on the other hand, indicated a small reduction in fold change values. When Ah and Et were infected, the fold change value of the CT treatment fell with no significance (*p* > 0.05). Following infection with Ah and Et, the values in T1 increased in an insignificant manner (*p* > 0.05). T2 exhibited a substantial (*p* < 0.05) rise in Ah, followed by a considerable (*p* < 0.05) drop in Et and, finally, an insignificant decline in T3 (*p* > 0.05).

### 3.7. Integrated Biomarker Responses Approach

#### 3.7.1. Gene Biomarkers

The IBR plot of treatments showed an interesting result among the treatments (Figure 5). In the first treatment group (A–D), the overall plot area in the feed trial can be represented as T2-90 > T3-90 > T1-90 > CT-90, and a similar trend was followed for a challenged study with *A. hydrophila* (E-H) and *E. tarda* (I-L). In the first treatment group, A showed the smallest and a similar coverage of the plot area for STAT1, and Mx and ISG 15 have similar coverages, as indicated with dots. B showed a greater area for STAT 1 and Mx, while STAT1 and Mx have almost the same area of the plot. C exhibited a maximum plot area for Mx, and ISG 15 and STAT 1 showed minor differences in the plot area. D showed a greater area for ISG 15, followed by Mx and STAT 1. In the second treatment group (E-H), more area was occupied by Mx, and a similar trend was followed by (I-L) as well. When the treatment was compared vertically, A showed a relatively greater area for STAT 1 than E and showed more area for Mx, D, and F, J showed more coverage for Mx, D and H depicted more area for ISG 15, and L represented almost similar areas for three genes.

#### 3.7.2. Treatment Biomarkers

When ISG 15, Mx, and STAT1 are considered as biomarkers, the trend was different (Figure 6a,b). In the feeding and challenge study, the shown experiment (CT-90) showed the maximum area, followed by T3-90 for ISG15. The same trend was followed by Mx, but STAT 1 showed the smallest area of coverage as compared with the two other genes.

## 4. Discussion

### 4.1. RNA Extraction, cDNA Synthesis, Semiquantitative PCR, Densiometric Analyses, Networking, and Correlation

The prime requisites of down-line molecular applications greatly depend on the quality of mRNA and cDNA rationalization. In the present experiment, the RNA gel showed two distinct bands that are an indication of the quality of RNA. The present study recorded a good quantity of RNA; that is why the expression, primer efficiency, amplification, and fold changes were distinct and lay down a standard of expression for a gene. The same observations were reported by Taylor et al. (2017) [22], who has summarized the important regulating factors of qPCR. The semiquantitative PCR is an intermediary salvage stage that confirms the presence of a gene with optimized PCR conditions so that further down-line analysis can be performed [23]; however, the prominence of a band may not guarantee a better gene expression, and vice versa [24,25]. The same observation was also recorded in the present study. The densiometric parameters such as Lane (%), Band (%), Rf, and RQ showed a diverse trend for three genes. The RQ and Rf are the indicators of the good quality and quantity of cDNA, and Band (%) and Lane (%) are considered as markers for the specificity of the genes. In the present study, in correlation plot of Mx and RQ showed a maximum correlation with Rf and Lane (%), while, in two other gens, RQ and Rf have a positive relationship with each other while other parameters differ, which might be due to gene specificity. The densitometry showed that a maximum value was recorded for ISG 15, while gene expression in the feeding experiment followed by the challenge study revealed a maximum expression for the Mx gene, which might be because an amplification of a particular gene tends to structural changes during PCR cycles. Similar observations were reported by Meena et al. (2021) [14], who have reported that better densitometry analyses in intact DNA samples during induced oxidative samples were being recovered with the application natural herbal oxidants. This gene-specific deviation is an indication of comparative densitometry plasticity.

### 4.2. qPCR

In the present study, three genes, namely, Mx, STAT1, and ISG15, have been used for testing the efficacy of herbal extracts in terms of gene modulation. The Mx is an antiviral protein belonging to the family of dynamins that includes an amino-terminal with a G-domain, an interactive central area, and a leucine zipper as an effector area of GED or GTPase [2]. Interferon is known to induce anti-proliferative, antiviral, and immunomodulatory proteins [26]. Out of three interferon-mediated proteins, the Mx proteins specially inhibit the protein synthesis of viruses including influenza and stomatitis [27]. The cellular functions of the Mx protein are still unclear; however, it is proven that all proteins have GTP binding site, of which the COOH-terminal leucine zipper domain is an important structural element [2].

STAT1 (Signal Transducer and Activator of Transcription) is an essential signal transduction protein that is intricate in the interferon pathway. STAT1 plays a significant role in the non-specific defense system [3]. ISG15 is an interferon-stimulating antiviral gene. Although ISG15 was thought to be an integral part of classical antiviral immunity, it has newly appeared as a regulator of genome steadiness, with main roles in the DNA nicking inhibition to modulate p53 signaling and error-free DNA replication [4]. The herbal materials or plants are mainly known to enhance non-specific immune systems; however, it has also been reported that they trigger the specific immune system, which in turn triggers the up-regulation of the immune genes, thereby protecting against pathogens. Similarly, Nhu et al. (2019) [28] evaluated the effects of five herbal extracts: garlic, neem, asthma-plant, bhumi amla, and ginger in *P. hypophthamus* fingerlings, which could enhance the specific immune parameters such as various types of cytokines including mhc class II cytokines. The previous study also showed that a very common spice garlic used as a food supplement has been recorded to enhance specific immune systems, i.e., rainbow trout [29], hybrid tilapia [30], Asian seabass [31], and Caspian roach [32].

The gene expression showed an elevated level of a respective gene upon infection, except for the control, which might be due to the immunomodulatory effects of the treatment at varying doses [33,34,35,36]. In the indoor feed trial followed by the challenge study, three genes, namely, Mx, ISG15, and STAT1, were expressed as the maximum expression of Mx. Most importantly, Mx was reported to express in the control treatment and in further infection with bacterial pathogens, which is also in agreement with Zavyalov et al. (2020) [37]. The other two genes, ISG15 and STAT1, exhibited no considerable (*p* > 0.05) expression in CT followed by infection, which might be due to the intrinsic nature of the availability of Mx in normal fish species but may not be the case for ISG15 and STAT1.

The same results were reported by Roy et al. (2016) [38] and Das et al. (2019) [39], who suggested that Mx acts as a natural gatekeeper of fish immune systems that protects them in early life stages and further gets activated when encountering the same pathogens. Additionally, the same results were found by other researchers, who reported that the expression of STAT1 is induced by Poly I:C, and upon infection, its expression was found to be significantly enhanced as compared to the control [3]. Upon infection, the expression of three genes was observed, but Mx showed its highest expression in the natural condition of fish, which is an indication that Mx did not activate directly due to the virus; rather, it is expressed with the activation of IFNS, while the other two genes are expressed only when they encounter pathogens. The same results were reported by Schiavano et al. (2016) [40]. In an indoor feed trial, Mx showed maximum expression as compared to the other two genes in all treatments, which might be due to the activation of IFNs due to herbal bioactive principles in the case of Mx, while it could be a weak or inappropriate factor in hitting the activation of IFNs in the other two genes, particularly when the fish was challenged with bacterial pathogens. The same observation was reported by previous researchers [41,42].

### 4.3. Integrated Biomarker Approach

Recently, there has been wide application of the IBR approach as a means to elucidate the stress perceived by fish due to various stress factors [16]. This approach could be utilized to elucidate the effects of culture aspects on water quality and fish growth [18]. The present study is an attempt to extend and link the dietary level of TABP-based feed on immunogenic gens in *L. rohita* to conclude the nutrigenomic interaction using an IBR approach. As reported in the present study, the value of gene biomarkers was higher at 1.0 g TABP-based feed as compared with other TABP biomarkers. This finding is in accordance with the qPCR gene expression. The gene biomarker showed a maximum IBR area value for the Mx gene in C, G, and K treatments; however, other genes showed different patterns, which might be due to the inherent availability of the Mx in fish. The same finding has also been recorded by Das et al. (2019) [39], who has reported the expression of the Mx gene in *L. rohita* in control fish as well. The expression of two other genes, ISG 15 and STAT 1, might be due to the higher optimum dietary TABP (1.5 g TABP) level that is required to trigger the molecular signaling for the expression of the genes. The value of treatment biomarkers showed a maximum value of CT.90 for ISG 15, CT.90, and T3.90 for Mx, and for STAT 1, all treatments biomarkers showed almost the same IBR value, which is likely due to the expression capability of the immunogenic bioactive principle of dietary TABP. The same finding was reported by Ahmadifar et al. (2021) [43], who has reported varying expression levels of immunogenic and growth genes in accordance with the dietary herbal additives. *E. Tarda* biomarkers showed a diverse trend, and T3.100Et showed a maximum IBR value for ISG 15 and showed the same IBR value for Mx and STAT 1. This deviation might be due to the specificity of a gene towards particular pathogens, which is witnessed with the diverse expression of genes upon the challenge with *A. hydrophila* and *E. tarda*.

## 5. Conclusions

The present study first revealed the immunogenic effects of dietary TABP in a fish model. A dose of 1.0 g·kg^−1^ TABP could enhance the expression of Mx, while ISG 15 and STAT 1 showed a dose-dependent gene response. The present study also validates the presence of the Mx gene in control fish (called the Natural Gate Keeper of fish immunity), which provides protection during early stages and in natural culture environments.

## Figures and Tables

**Figure 1 animals-13-00039-f001:**
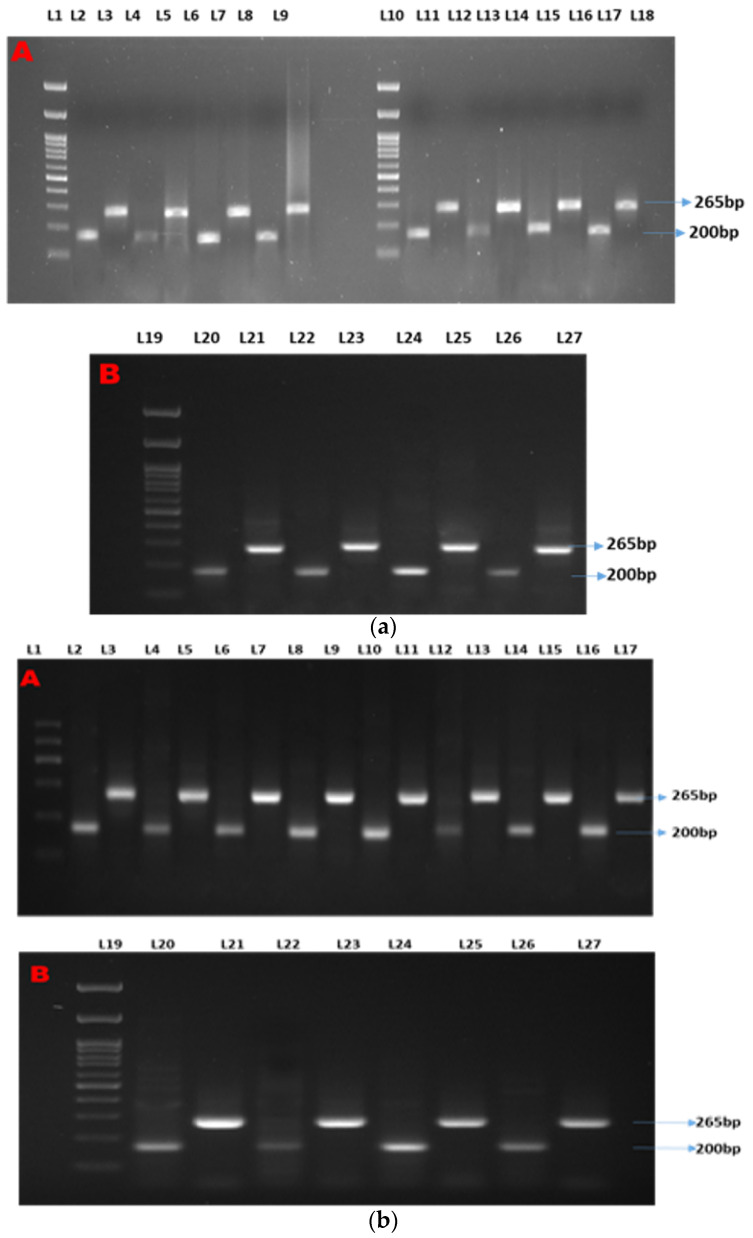
(**a**–**c**) show the semi-quantitative PCR of targeted genes and housekeeping genes. Legends. (**a**) Showing semi-quantitative PCR of ISG15. (**b**) Showing semi-quantitative PCR of Mx gene. (**c**) Showing semi-quantitative PCR of STAT1 gene. L-1, 11, and 19: 1Kb molecular marker; L-2, 4, 6, 8, 11,13, 15, 17, 20, 22, 24, and 26: β-actin (housekeeping gene); L-3: CT90; L-5: T1-90; L-7: T2-90; L-9: T3-90; L-12: CT100-Ah; L-14: T1-100Ah; L-16: T2-100Ah; L-18: T3-100Ah; L-21: CT100-Et; L-23: T1-100Et; L-25: T2-100Et; L-27: T3-100Et.

**Figure 2 animals-13-00039-f002:**
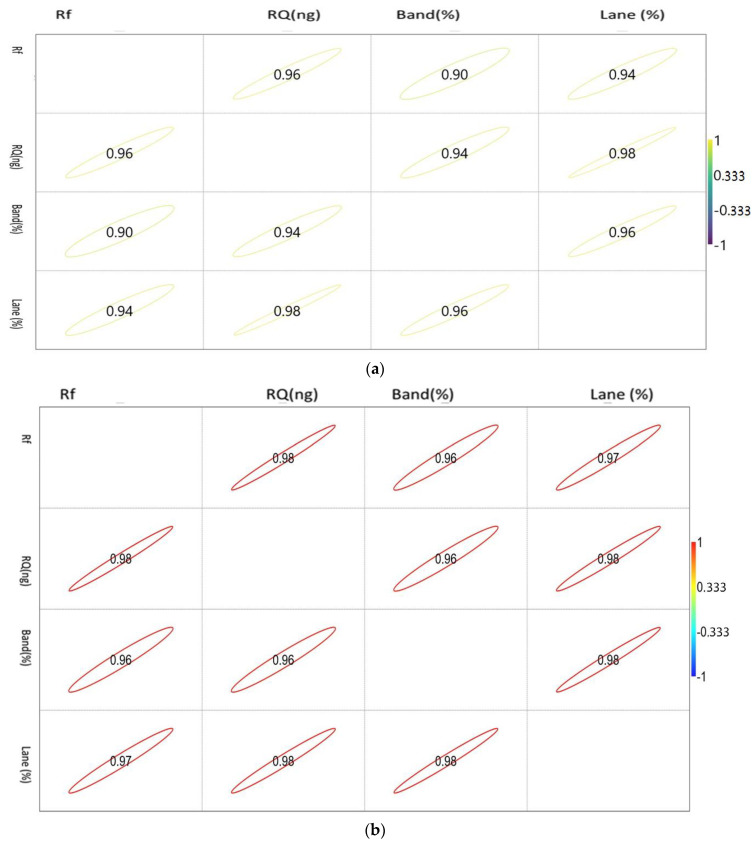
(**a**). Showing the correlation between the densiometric parameters of the Mx gene. (**b**). Showing the correlation between the densiometric parameters of the ISG15 gene. (**c**). Showing the correlation between the densiometric parameters of the STAT1 gene.

**Figure 3 animals-13-00039-f003:**
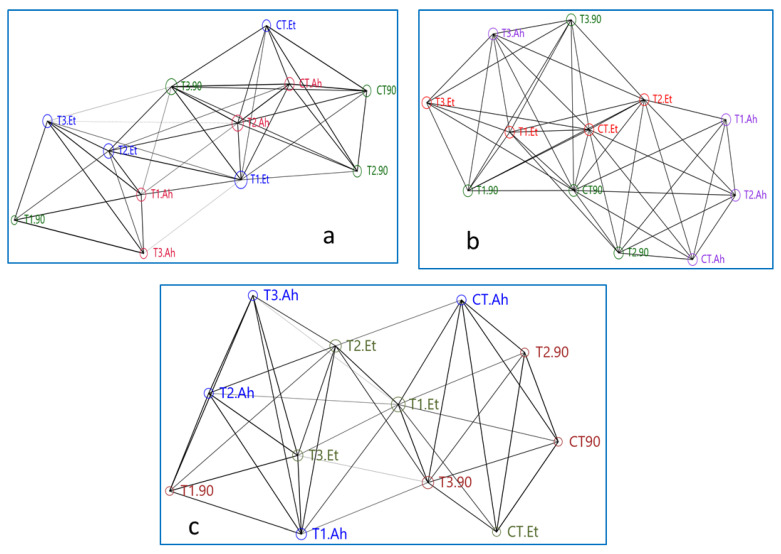
(**a**–**c**) Showing networking among treatment groups based on genes densiometric parameters. Here, treatment groups followed by a challenge study are being represented as follows: CT, T1, T2, and T3 are the treatment groups, suffix-90 indicates a 90-day feeding trial of these treatment groups, and Et and Ah represent the bacterial isolates *Edwardsiella tarda* and *Aeromonas hydrophila*, respectively.

**Figure 4 animals-13-00039-f004:**
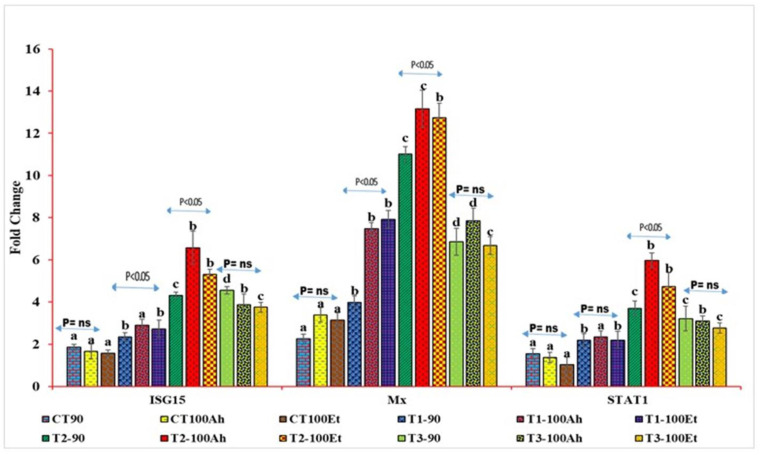
qPCR analyses of three immunogenic genes (Mx, STAT1, and ISG15) during the feeding trial followed by the challenge study by two bacterial isolates, *Edwardsiella tarda* and *Aeromonas hydrophila.* Here, we conducted statistical analysis between and within the treatments. The superscript above the same color column shows a comparison between the treatments, and significance with an arrow (*p* < 0.05 representing significance at 5 % level of significance and ns represents insignificance) shows comparison among the treatments. CT, T1, T2, and T3 suffixed with 90 shows the feed trial of 90 days with treatment groups, and 100Ah and 100Et shows the challenge study with bacterial isolates, *Edwardsiella tarda* and *Aeromonas hydrophila*, for 10 days (90–100 days).

**Figure 5 animals-13-00039-f005:**
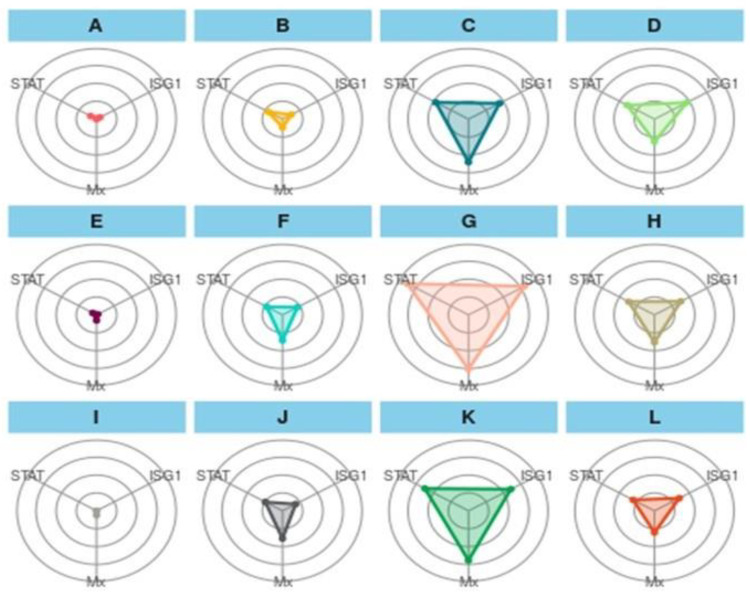
Integrated biomarker response approach plot of gene biomarkers in a feeding trial followed by a challenge study with two bacterial isolates, *Edwardsiella tarda* and *Aeromonas hydrophila*, for 10 days (90–100 days). Here, (**A**): CT-90-control treatment at 90 days; (**B**): T1-90-T1 treatment at 90 days; (**C**): T2-90-T2 treatment at 90 days; (**D**): T3-90-T3 treatment at 90 days; (**E**): CT-Ah-Control fish challenge with *A. hydrophila* at 100 days; (**F**): T1-Ah-T1 fish challenge with *A. hydrophila* at 100 days; (**G**): T2-Ah-T2 fish challenge with *A. hydrophila* at 100 days; (**H**): T3-Ah-T3 fish challenge with *A. hydrophila* at 100 days; (**I**): CT-Et-Control fish challenge with *E. tarda* at 100 days; (**J**): T1-Et-T1 fish Control fish challenge with *E. tarda* at 100 days; (**K**): T2-Et-T2 fish challenge with *E.Tarda* at 100 days; (**L**): T3-Et-T3 fish challenge with *E. tarda* at 100 days. STAT: STAT1 gene; ISG1: ISG 15.

**Figure 6 animals-13-00039-f006:**
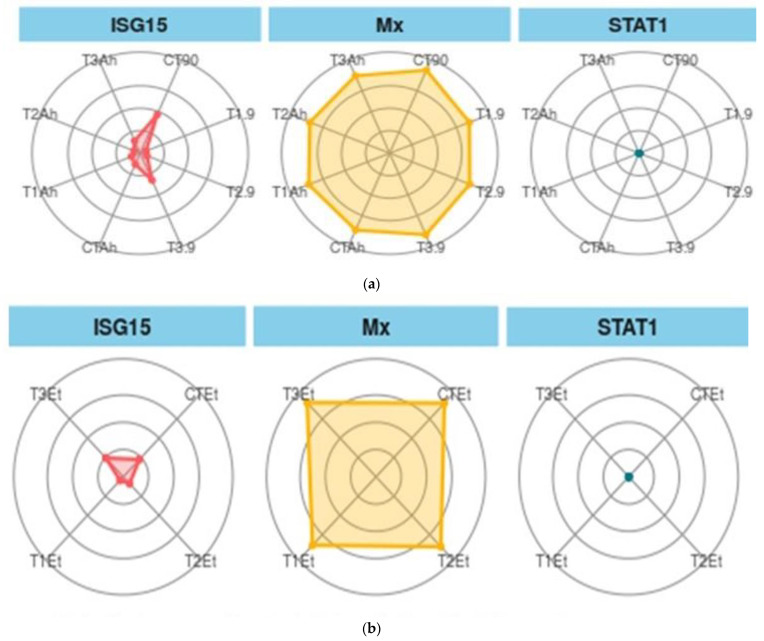
(**a**). Treatment biomarkers in the feeding trial followed by the challenge study. (**b**). Treatment biomarkers in the feeding trial followed by the challenge study. Here, CT, T1, T2, and T3 are the treatment groups, Mx, ISG15, and STAT1 are the targeted immunogenic genes, the suffix with 90 shows the feeding trial duration of 90 days, and the suffixes with Et and Ah represent the challenge study with two bacterial isolates, *Ewardsiella tarda* and *Aeromonas hydrophila*, for 10 days (90–100 days).

**Table 1 animals-13-00039-t001:** Details of the Primers used in the Present Study.

Primer	Forward	Reverse	Accession Number
Mx	5′-GTCCAGTACCACATGCTGGACC	5′-TTTGCCAGCACTCCTCAGGCGT-3′	KM216417
ISG15	5′-GGCAAAAGATCGTGTCTCGT-3′	5′-CATCACGGCATTGAAAACA-3′	KP604219
STAT1	5′-AGAAGGGCCAGGTCAAAACT-3′	5′-TCCACAGCCAGAATGGTACA-3′	Kept on hold for the complete sequence
β-actin	5′-TTCGAGCAGGAGATGGGCACTG- 3′	5′-GCATCCTGTCAGCAATGCCA-3′	Housekeeping gene (EU184877)

**Table 2 animals-13-00039-t002:** Showing densitometry parameters of the Mx gene.

Trt	Rf	RQ (ng)	Band (%)	Lane (%)
** β-actin	0.94–0.98	1.38–1.46	52.8–59.35	32.77–38.31
CT.90	0.73 ± 0.12 ^a^	1.12 ± 0.12 ^a^	35.63 ± 1.17 ^a^	22.18 ± 2.17 ^a^
T1.90	0.74 ± 0.07 ^b^	1.17 ± 0.17 ^ab^	37.42 ± 0.96 ^a^	26.56 ± 1.08 ^abc^
T2.90	0.83 ± 0.06 ^b^	1.29 ± 0.21 ^cd^	47.56 ± 1.53 ^d^	29.43 ± 1.48 ^bcde^
T3.90	0.89 ± 0.06 ^b^	1.34 ± 0.26 ^d^	49.88 ± 1.18 ^d^	32.33 ± 2.19 ^fg^
CT.100Ah	0.79 ± 0.02 ^b^	1.14 ± 0.28 ^a^	38.43 ± 1.29 ^ab^	24.45 ± 2.94 ^ab^
T1.100Ah	0.83 ± 0.08 ^b^	1.26 ± 0.31 ^c^	41.27 ± 2.13 ^bc^	28.42 ± 1.87 ^cdef^
T2.100Ah	0.98 ± 0.05 ^c^	1.46 ± 0.35 ^e^	59.35 ± 3.21 ^e^	38.31 ± 3.28 ^h^
T3.100Ah	0.95 ± 0.04 ^c^	1.42 ± 0.18 ^e^	49.23 ± 2.38 ^d^	34.68 ± 2.19 ^gh^
CT.100Et	0.78 ± 0.08 ^b^	1.12 ± 0.24 ^a^	37.23 ± 1.78 ^ab^	23.31 ± 2.18 ^abc^
T1.100Et	0.81 ± 0.02 ^b^	1.22 ± 0.38 ^bc^	40.12 ± 0.59 ^bc^	26.35 ± 1.84 ^bcd^
T2.100Et	0.93 ± 0.05 ^c^	1.34 ± 0.16 ^d^	46.35 ± 1.67 ^d^	31.26 ± 2.72 ^efg^
T3.100Et	0.87 ± 0.07 ^c^	1.31 ± 0.26 ^d^	43.37 ± 0.79 ^c^	30.18 ± 2.18 ^def^

Data represent n = 3. The superscript in the same row differs significantly (*p* < 0.05); ** represents the densitometry parameters value of β-actin as a positive control for all three trials (feed trial, challenge study with *A. hydrophila* and *E. tarda*).

**Table 3 animals-13-00039-t003:** Showing densitometry parameters of the ISG 15 gene.

Trt	Rf	RQ (ng)	Band (%)	Lane (%)
** β-actin	0.95–0.99	1.37–1.49	53.4–59.49	33.05–39.67
CT.90	0.77 ± 0.08 ^a^	1.18 ± 0.18 ^a^	37.73 ± 3.78 ^abc^	25.98 ± 2.52 ^a^
T1.90	0.71 ± 0.07 ^a^	1.14 ± 0.21 ^a^	34.32 ± 2.67 ^ab^	24.34 ± 1.57 ^a^
T2.90	0.87 ± 0.04 ^cd^	1.32 ± 0.14 ^bc^	49.78 ± 3.28 ^bcd^	33.42 ± 2.81 ^bc^
T3.90	0.84 ± 0.02 ^bc^	1.31 ± 0.18 ^bc^	46.56 ± 2.53 ^a^	32.78 ± 3.81 ^bc^
CT.100Ah	0.88 ± 0.07 ^cd^	1.34 ± 0.26 ^e^	51.75 ± 3.92 ^cd^	34.56 ± 3.29 ^c^
T1.100Ah	0.81 ± 0.04 ^ab^	1.29 ± 0.23 ^b^	47.27 ± 2.86 ^bcd^	31.23 ± 2.45 ^b^
T2.100Ah	0.99 ± 0.05 ^f^	1.49 ± 0.17 ^g^	59.49 ± 2.76 ^d^	39.67 ± 1.98 ^g^
T3.100Ah	0.95 ± 0.06 ^ef^	1.45 ± 0.18 ^f^	53.54 ± 3.89 ^d^	37.86 ± 2.65 ^f^
CT.100Et	0.82 ± 0.04 ^ab^	1.3 ± 0.24 ^cd^	46.52 ± 3.56 ^bcd^	32.23 ± 2.15 ^bc^
T1.100Et	0.84 ± 0.04 ^bc^	1.32 ± 0.31 ^cde^	46.73 ± 2.15 ^bcd^	32.65 ± 1.87 ^bc^
T2.100Et	0.93 ± 0.07 ^ef^	1.44 ± 0.17 ^fg^	52.23 ± 3.19 ^d^	35.65 ± 1.54 ^d^
T3.100Et	0.89 ± 0.03 ^d^	1.36 ± 0.19 ^de^	49.89 ± 2.58 ^bcd^	35.63 ± 2.14 ^de^

Data represent n = 3. The superscript in the same row differs significantly (*p* < 0.05); ** represents the densitometry parameters value of β-actin as a positive control for all three trials (feed trail, challenge study with *A. hydrophila* and *E. tarda*).

**Table 4 animals-13-00039-t004:** Showing densitometry parameters of the STAT 1 gene.

Trt	Rf	RQ (ng)	Band (%)	Lane (%)
** β-actin	0.92–0.96	1.33–1.41	50.8–57.35	31.77–35.31
CT.90	0.73 ± 0.06 ^ab^	1.12 ± 0.32 ^a^	35.63 ± 4.23 ^a^	22.18 ± 3.95 ^a^
T1.90	0.74 ± 0.07 ^a^	1.17 ± 0.23 ^a^	37.42 ± 3.67 ^ab^	26.56 ± 1.23 ^bc^
T2.90	0.83 ± 0.04 ^cd^	1.29 ± 0.14 ^bc^	47.56 ± 2.17 ^ef^	29.43 ± 1.95 ^de^
T3.90	0.89 ± 0.03 ^ef^	1.34 ± 0.12 ^c^	49.88 ± 3.21 ^fg^	32.33 ± 2.85 ^efg^
CT.100Ah	0.79 ± 0.08 ^bc^	1.14 ± 0.18 ^a^	38.43 ± 2.85 ^b^	24.45 ± 3.18 ^bc^
T1.100Ah	0.83 ± 0.05 ^cd^	1.26 ± 0.27 ^b^	41.27 ± 2.96 ^c^	28.42 ± 2.63 ^cd^
T2.100Ah	0.98 ± 0.08 ^g^	1.44 ± 0.16 ^d^	52.37 ± 3.98 ^g^	36.57 ± 3.21 ^fg^
T3.100Ah	0.95 ± 0.05 ^fg^	1.42 ± 0.20 ^a^	49.23 ± 2.69 ^fg^	34.68 ± 2.96 ^g^
CT.100Et	0.78 ± 0.07 ^ab^	1.12 ± 0.34 ^a^	37.23 ± 3.78 ^ab^	23.31 ± 3.82 ^ab^
T1.100Et	0.81 ± 0.08 ^cd^	1.22 ± 0.16 ^b^	40.12 ± 2.82 ^bc^	26.35 ± 2.94 ^bc^
T2.100Et	0.93 ± 0.08 ^ef^	1.34 ± 0.18 ^c^	46.35 ± 2.92 ^de^	31.26 ± 1.96 ^ef^
T3.100Et	0.87 ± 0.03 ^de^	1.31 ± 0.02 ^c^	43.37 ± 3.62 ^c^	30.18 ± 2.47 ^e^

Data represent n = 3. The superscript in the same row differs significantly (*p* < 0.05); ** represents the densitometry parameters value of β-actin as a positive control for all three trials (feed trail, challenge study with *A. hydrophila* and *E. tarda*).

## Data Availability

The data will be made available based on the request.

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
