# Peer review of "Immunogenic Effects of Dietary Terminalia arjuna Bark Powder in Labeo rohita, a Fish Model: Elucidated by an Integrated Biomarker Response Approach"

_animals, 2022, doi:10.3390/ani13010039_

Round 1
Reviewer 1 Report
General, the article needs basic spell check and check for colloquial English
Line 12. Intensive aquaculture is never unmanaged. Suggest removing unmanaged.
Line 13. Comm= common. This also needs further scope. Western aquaculture uses little to no antibiotics. Vaccination is the most common remedy there. The authors need to give more explanation in which part of aquaculture this is the case
Line 14 remove a before havoc, suggest rephrasing this to more scientific language
Line 15, remove The.
Line 16 Replace obnoxious chemicals for medicine or undesired medicine. Refer to one health.
Line 16-17. Rephrase whole sentence. Not proper English
Line 18 add ‘the’ between evaluate and medicinal
Line 19 into= in
12.3g/kg was not a group in the study so can’t be a recommendation. There is also no further justification shown for this anywhere else in the paper
Line 24 created=produced/manufactured
Line 42. Plural of carp is carp. Suggest saying ‘major carp species’
Line 44. Full stop missing
Line 45. Authors introduce abbreviation for India Major carp species but not using it. Carp with no s, remove ‘like
Line 46 multiple spaces after with
Line 51 ‘mainly due to bacteria’
Line 52. ‘Flavobacterium ‘ check spelling. Also, what species?? If multiple it is common to say ‘Flavobacterium spp.
Line53 what is meant with increased number of bacteria?
Number of species of bacteria infecting carp
Bacteria infecting individual fish
Please rephrase
Line 54. ‘innate’ (spelling). Why no swith to viral infection for immune system. Author is discussion bacteria first and now discussing immune system for viral infection. No viral infection are mentioned so not appropriate to say here. Also why not discuss innate immune response to bacteria instead.
Remove all mention of viral infections. Challenge model is with bacteria. Rewrite this section line 54-68
Line 71-72 include references
Line 81 , remove The fish species including IMCs.
Rewrite section 81-89. Makes no sense and has duplication
Stating that vaccination is expensive is incorrect. First of all ‘expensive’ is an opinion not a fact. Economic viability is better term to use. There are plenty studies for economic return on vaccines. It may be believed it is but would be good to point out that that may not be actually true
Line 101-113, what is the relevance of this to the study.? Remove. You are introducing a new product to the production of carp. It does not make it circular economy. Even if it is a green product.
Line 114 indices, Indicator??
Note:
Section 2.1 not detailed enough. Include
1. Group design for feed trial
2. Number of replicates
3. Bacterial species used for infection model
a. Growth conditions
b. Titre
c. Route of infection
4. Infection model
5. Experimental design
6. Tank set up (flow through, recirculation etc)
A reviewer/reader needs to heave enough information to recreate the study which not the case currently
Section 2.4. biomakers were introduced as antiviral, yet you use them on bacterial infection. This is in appropriate
The authors introduce the biomakers used as viral makers, but use a bacterial model to evaluate the (as that is the main disease challenge in the real world). Some of these biomakers are involved in bacterial infections but not all in the same way. Although, the study may be sound it is currently not written up in an appropriate manor to conclude that. I also request that all studies involving animals have an ethical approval number stated.
Due to the flaws present I have not continued the evaluation.
Author Response
Reviewer 1
Q1:General, the article needs basic spell check and check for colloquial English
R1: The article has been revised for spell check and colloquial English.
Q2:Line 12. Intensive aquaculture is never unmanaged. Suggest removing unmanaged.
R2: corrected as suggested
Q3:Line 13. Comm= common. This also needs further scope. Western aquaculture uses little to no antibiotics. Vaccination is the most common remedy there. The authors need to give more explanation in which part of aquaculture this is the case
R3: The text modified as suggested to make it more clear, thank you.
Q4:Line 14 remove a before havoc, suggest rephrasing this to more scientific language
R4: The sentence has been changed.
Q5:Line 15, remove The.
R5: The has been removed
Q6:Line 16 Replace obnoxious chemicals for medicine or undesired medicine. Refer to one health.
R6: The correction has been made as suggested, Thank you.
Q7:Line 16-17. Rephrase whole sentence. Not proper English
R7: The correction has been made as per suggestion.
Q8:Line 18 add ‘the’ between evaluate and medicinal
R8: The correction has done in light of comments.
Q9:Line 19 into= in
12.3g/kg was not a group in the study so can’t be a recommendation. There is also no further justification shown for this anywhere else in the paper.
R9: Yes, you are correct and now it has mention along with the data in text itself. It is optimised dose retrieved by the broken line regression. The article has been published with following citation.
D.K. Meena, A.K. Sahoo, M. Jayant, N.P. Sahu, P.P. Srivastava, H.S. Swain, B.K. Behera, K. Satvik, B.K. Das, Bioconversion of Terminalia arjuna bark powder into a herbal feed for Labeo rohita: Can it be a sustainability paradigm for Green Fish production?, Animal Feed Science and Technology,Volume 284,2022, 115132.
Q10: Line 24 created=produced/manufactured
R10: he sentence has been changed as per the suggestion.
Q11:Line 42. Plural of carp is carp. Suggest saying ‘major carp species’
R11:Correction has been made as suggested.
Q12:Line 44. Full stop missing
R12:The full stop has put as mentioned, thank you.
Q13:Line 45. Authors introduce abbreviation for India Major carp species but not using it. Carp with no s, remove ‘like
R13: The correction has been made as suggested.
Q14:Line 46 multiple spaces after with
R14: The spaces has beed reduced as pointed.
Q15:Line 51 ‘mainly due to bacteria’
R15: The sentence has been rephrased for better clarification, thank you.
Q16:Line 52. ‘Flavobacterium ‘ check spelling. Also, what species?? If multiple it is common to say ‘Flavobacterium spp.
R16:The spelling has been corrected as highlighted.
Q17”Line53 what is meant with increased number of bacteria?
Number of species of bacteria infecting carp
Bacteria infecting individual fish
R17: it is meant for Number of species of bacteria infecting carp
Q18:Please rephrase
Line 54. ‘innate’ (spelling). Why no swith to viral infection for immune system. Author is discussion bacteria first and now discussing immune system for viral infection. No viral infection are mentioned so not appropriate to say here. Also why not discuss innate immune response to bacteria instead.
Remove all mention of viral infections. Challenge model is with bacteria. Rewrite this section line 54-68.
R18: Yes correctly said that the study has used bacterial model for infection but taking reference from previous studies the same gene have been also used to study the impact of herbal product on fish adaptive immune system which is very much vital from fish helath management point of view.
The purpose of this study was to evaluate the immunomodulatory effects of three genes. Out of them Mx is consider as natural gate keeper of the fish immune system. So purpose of this study was to
1.Evaluate the Dietary TABP based immunomodulatory effects of these three genes.
- STST1 gene is firstly was found to express in L.rohita as mentioned in text also.
Q19:Line 71-72 include references
R19: The sentence has been made simple and need not to have the reference.
Q20:Line 81 , remove The fish species including IMCs.
R20: The text has been deleted s suggested.
Q21:Rewrite section 81-89. Makes no sense and has duplication
R21: The section has been deleted as it seems duplication to earlier text, thank you.
Q22:Stating that vaccination is expensive is incorrect. First of all ‘expensive’ is an opinion not a fact. Economic viability is better term to use. There are plenty studies for economic return on vaccines. It may be believed it is but would be good to point out that that may not be actually true.
R22:The text has been modified accordingly, Thank you.
Q23:Line 101-113, what is the relevance of this to the study.? Remove. You are introducing a new product to the production of carp. It does not make it circular economy. Even if it is a green product.
R23:Th section has ben removed as per the suggestion.
Q24:Line 114 indices, Indicator??
R24:It is indices
Q25:Note:
Section 2.1 not detailed enough. Include
- Group design for feed trial
- Number of replicates
- Bacterial species used for infection model
- Growth conditions
- Titre
- Route of infection
- Infection model
- Experimental design
- Tank set up (flow through, recirculation etc)
A reviewer/reader needs to heave enough information to recreate the study which not the case currently.
R25:The material and methods section include most of the things in section 2.1 however left over details are mentioned in the section, Thank you.
The rearing size of fish was selected because this stage onward they are being used as stocking materials and transferred to the new environments and thereby there is very much possibility to occur infection so in view this this the size of fish was selected. The area of cultured one fish is kept upper side and it was taken 10 L for one fish and total volume is 450 L out in500 L capacity tank.
The experiment was conducted for three months (90 days) as cultured period for rearing stage of fish and to provide a suitable time to reflect dietary effects of TABP on fish specific immune system unlike other herbal materials. Further, this study followed by challenge study with two pathogenic bacterial isolates that was lasted up to 10 days.
Thank for your points but it’s not like that we have every information but this part we have published already and for which we will put refence and to limit the number of table and figure we have not included that information by as per the comments we will include all the details. The below is the article published which contains all these information’s.
D.K. Meena, A.K. Sahoo, M. Jayant, N.P. Sahu, P.P. Srivastava, H.S. Swain, B.K. Behera, K. Satvik, B.K. Das, Bioconversion of Terminalia arjuna bark powder into a herbal feed for Labeo rohita: Can it be a sustainability paradigm for Green Fish production? Animal Feed Science and Technology,Volume 284,2022, 115132.
D.K. Meena, N.P. Sahu, P.P. Srivastava, M. Jadhav, R. Prasad, R.C. Mallick, A.K. Sahoo, B.K. Behera, D. Mohanty, B.K. Das. Effective valorisation of facile extract matrix of Terminalia arjuna (Roxb) against elite microbes of aquaculture industry–a credence to bioactive principles: Can it be a sustainability paradigm in designing broad spectrum antimicrobials? Industrial Crops and Products, 171,2021,113905.
Q26: Section 2.4. biomakers were introduced as antiviral, yet you use them on bacterial infection. This is in appropriate
The authors introduce the biomakers used as viral makers, but use a bacterial model to evaluate the (as that is the main disease challenge in the real world). Some of these biomakers are involved in bacterial infections but not all in the same way. Although, the study may be sound it is currently not written up in an appropriate manor to conclude that. I also request that all studies involving animals have an ethical approval number stated.
R26:yes its true but you might be aware that those biomarker are responsible for disease protection by other factors also and particularly MX1 is therefore of called as natural gate keeper. The presence of Mx in Labeo rohita is already established by Das et al. (2016). Present study is of its first kind to evaluate the expression of STAT1 in Labeo rohita and ISG 15 also expressed consequent upon the bacterial infection. So present study is designed to see whether those important genes can be modulated due to the applications of T.arjuna based fish feed. The following is he ciation of previous research
Das BK, Roy P, Rout AK, Sahoo DR, Panda SP, Pattanaik S, Dehury B, Behera BK, Mishra SS. Molecular cloning, GTP recognition mechanism and tissue-specific expression profiling of myxovirus resistance (Mx) protein in Labeo rohita (Hamilton) after Poly I:C induction. Scientific Reports (2019) 9: doi: 10.1038/s41598-019-40323-0.
Q27: Due to the flaws present I have not continued the evaluation.
R27: With due respect, we have revised the whole manuscript and now it is in shape for publication. Thank you.

Reviewer 2 Report
The main question in this research was that the selected genes was antiviral related genes, but Terminalia arjuna in this study was mainly antibacterial properties, and challenge study was conducted using two bacteria. I think the topic is somewhat original in the field, and also addresses a specific gap. Methods: This study provided an integrated biomarker response approach as reliable marker to evaluate the impact of multiple drivers in holistic manner. To be honest, I don't fully understand this approach, but I think this innovation should be encouraged. Integrated Biomarker Approach should be described in more detail so that the reader can easily understand and repeat it. Conclusion: The conclusions are consistent with the evidence and arguments presented and they address the main question posed. The references are appropriate. Tables/Figures: Perhaps some clearer figures, such as figure 2a, ab and 2c, could be provided.
More comments:
L29 This phrase “myxovirus resistance gene” needs to be followed by brackets.
L32 The first time IBR was mentioned in the abstract. Please provide the full name.
L88-89 Since the author studies the expression of antiviral related genes, it is necessary to introduce in detail which viral diseases seriously affect the production of Labeo rohita.
L128-129 Why did the authors choose antiviral related genes and then challenge study with bacteria?
L135 Please provide the details of bacteria and the process of challenge study with two bacteria.
L159 In table 1, please provider the NCBI No. of the genes.
5. Conclusion
“present study also validates ...” “present study” should be revised to “the present study”.
Author Response
Reviewer 2
Q28:L29 This phrase “myxovirus resistance gene” needs to be followed by brackets.
R28: The correction has been mad asp per suggestion
Q30:L32 The first time IBR was mentioned in the abstract. Please provide the full name.
R30: The full form of IBR has been provided in abstract. Thank you
Q31:L88-89 Since the author studies the expression of antiviral related genes, it is necessary to introduce in detail which viral diseases seriously affect the production of Labeo rohita.
R31: The purpose of studying the targeted gene is different hearer as they are natural gate keepers for the fish during early states and some of them has already studied by other researchers in carps which but induced by different feed additives or bioactive compounds.
Q32:L128-129 Why did the authors choose antiviral related genes and then challenge study with bacteria?
R32: These gene are now being used a marker for gene expression because they are supposed to express during bacterial infection also and in present study the aim is to evaluate whether the Terminalia arjuna also has any impeach on gene level unlike other herbal medicament. Second out of tree genes have already studied in carps but induced by different agents and challenged wit bacterial strains. Since bacteria are significant causative agents in carps and account a total of 20-30 % monetary loss. Therefore, we used these markers gene which can be used as standard broad-spectrum markers for gene expression and per se activity during bacterial infection. A far as the esteemed reviewers comments the virus has also a significant loss in carps and below are some of the references.
Pradhan, P.,Paria, Anutosh, Yadav, Manoj, Verma, Dev, Gupta, Shubham, Swaminathan, T Raja, Rathore, Gaurav, Sood, Neeraj , Lal, Kuldeep. (2019). Susceptibility of Indian major carp Labeo rohita to tilapia lake virus. Aquaculture. 515. 734567. 10.1016/j.aquaculture.2019.734567.
Q33:L135 Please provide the details of bacteria and the process of challenge study with two bacteria.
R33: The details and process of challenge of bacteria has been provided.
Q34:L159 In table 1, please provider the NCBI No. of the genes.
R34:The table has been completed as feasible.
- Conclusion
Q35:“present study also validates ...” “present study” should be revised to “the present study”.
R35: The correction has been done as suggested, Thank you.
Q36: The Tables/Figures: Perhaps some clearer figures, such as figure 2a, ab and 2c, could be provided.
R37:The figure 2a, 2b, and @c has been modified up to extent possible as it was made in PAST software so that cannot change much . I tried my best to make clear for better representation. Thank you.

Round 2
Reviewer 2 Report
No